# Transparent Alicyclic Polyimides Prepared via Copolymerization or Crosslinking: Enhanced Flexibility and Optical Properties for Flexible Display Cover Windows

**DOI:** 10.3390/polym17152081

**Published:** 2025-07-30

**Authors:** Hyuck-Jin Kwon, Jun Hwang, Suk-Min Hong, Chil Won Lee

**Affiliations:** 1Department of Chemistry, College of Science and Technology, Dankook University, Cheonan-si 31116, Republic of Korea; 12190616@dankook.ac.kr (H.-J.K.); 12190615@dankook.ac.kr (S.-M.H.); 2Department of Foundry Engineering, Dankook University, Yongin-si 16890, Republic of Korea; abc@dankook.ac.kr

**Keywords:** transparent polyimides, flexibility, optical transmittance, refractive index, co-polymerization, crosslinking

## Abstract

Transparent polyimides with excellent mechanical properties and high optical transmittance have been widely used in various optical and electrical applications. However, due to the rigidity of their aromatic structure, their flexibility is limited, making them unsuitable for applications requiring different form factors, such as flexible display cover windows. Furthermore, the refractive index of most transparent polyimides is approximately 1.57, which differs from that of the optically clear adhesives (OCAs) and window materials that have values typically around 1.5, resulting in visual distortion. This study employed 4,4′-(hexafluoroisopropylidene)diphthalic anhydride (6FDA) and 2,2′-bis(trifluoromethyl)benzidine (TFMB) as the base structure of polyimides (6T). Additionally, 1,3-bis(aminomethyl)cyclohexane (BAC) with a monocyclic structure and bis(aminomethyl)bicyclo[2,2,1]heptane (BBH) with a bicyclic structure were introduced as co-monomers or crosslinking agents to 6T. The mechanical, thermal, and optical properties of the obtained copolymers (6T-BAC and 6T-BBH series) and crosslinked polymers (6T-CL-BAC and 6T-CL-BBH series) were compared. Both the copolymer series (6T-BAC and 6T-BBH) and the crosslinked series (6T-CL-BAC and 6T-CL-BBH) exhibited improved optical properties compared to the conventional 6T, with maximum transmittance exceeding 90% and refractive indices ranging from approximately 1.53 to 1.55. Notably, the copolymer series achieved transmittance levels above 95% and exhibited lower refractive indices (~1.53), demonstrating superior optical performance relative not only to the 6T baseline but also to the crosslinked series. The alicyclic polyimides synthesized in this study exhibited mechanical flexibility, high optical transmittance, and a refractive index approaching 1.5, demonstrating their applicability for use as flexible display cover window materials.

## 1. Introduction

Display cover windows refer to the protective layers placed over the screens of electronic devices, such as smartphones, tablets, laptops, or wearable devices. The traditional materials used for display cover windows are tempered glass, which offers abrasion resistance and aesthetics. However, it is vulnerable to impact and lacks flexibility, making it unsuitable for cover windows that require high curvature [1,2,3]. To address this, ultra-thin tempered glass (UTG) has been developed for flexible and foldable cover windows [4,5,6]. Due to its inherent brittleness, UTG is difficult to process and handle during manufacturing, and it has limitations in achieving high curvature, highlighting the need for alternative materials [7,8,9,10].

The material used for cover windows in high-curvature displays must meet two important requirements. First, it must have high mechanical durability and flexibility to prevent surface cracks or delamination caused by repeated folding and bending [11,12]. Secondly, it should have high light transmittance and a low refractive index to reduce light reflection and prevent field distortion [13,14,15]. If the cover window has a high refractive index, reflectance increases according to the Fresnel equations, and this reflection can reduce display readability in strong lighting or outdoor environments, potentially causing visual distortion [16,17,18,19]. Additionally, the cover window includes an optically clear adhesive (OCA) or optically clear resin (OCR) for binding with the display panel and between the layers of the cover window [20,21], and most commercial OCA and OCR have a refractive index of around 1.5. A difference in the refractive index between OCA, OCR, and the window material can cause light refraction and reflection, potentially leading to image distortion and affecting the user’s field of view [22,23]. As a result, for the flexible display cover window to maintain high optical performance, it is essential to ensure that the material has a refractive index of approximately 1.5.

To meet these requirements, research on various alternative materials has been actively conducted, with polyimides emerging as the most promising candidate [10,24]. Polyimide is an engineering plastic widely used in aerospace applications, electronics, displays, and semiconductors [25,26]. Its high-temperature stability and excellent chemical resistance, mechanical strength, and electrical insulation properties make it a key insulating material in flexible electronic devices and high-density communication systems [27,28,29]. Unlike conventional tempered glass, polyimide is not brittle and can be bent and folded without cracking, providing the high durability and flexibility required for display cover windows. However, the refractive index of typical polyimide is relatively high (approximately 1.7), and its charge transfer complex (CTC) structure always generates yellow or brown colors [30,31]. Additionally, while the rigid backbone based on aromatic compounds can achieve high mechanical strength, polyimide tends to be less flexible, leading to issues such as crease marks and dents. These limitations restrict the applications of polyimides as display cover windows for high-curvature displays [32].

Various methods have been developed to improve the flexibility of polyimides. Previous studies show that using diamine or dianhydride with amide or ether bonds can maintain rigidity while increasing the fluidity of the molecular chains, ultimately enhancing the flexibility of polyimides [32,33]. By introducing monomers with an alicyclic structure, it is possible to increase the free volume, reduce rigidity, and achieve flexibility. Moreover, the non-conjugated alicyclic structure reduces the π electron density and lowers the refractive index. In comparison, the refractive index can be controlled by dispersing nanoparticles such as SiO_2_ and adjusting the size and dispersion of the particles. However, when the added nanoparticles are not evenly distributed in the matrix, they can reduce optical uniformity and degrade mechanical properties, leading to brittleness. Therefore, modifying the molecular structure of polyimide is the most effective way to adjust the refractive index while maintaining its physical properties.

This study synthesized polyimides with high flexibility and a refractive index of 1.5 by introducing alicyclic diamines into the transparent polyimides. First, hexafluoroisopropylidene)diphthalic anhydride (6FDA) and 2,2′-bis(trifluoromethyl)benzidine (TFMB) were used as the base structures of the polyimides (6T). Next, 1,3-bis(aminomethyl)cyclohexane (BAC) with a monocyclic structure and bis(aminomethyl)bicyclo[2,2,1]heptane (BBH) with a bicyclic structure were introduced as co-monomers or crosslinking agents with respect to 6T. Specifically, 6T-BAC and 6T-BBH copolymers were prepared by partially replacing TFMB with BAC and BBH, respectively. As the amount of alicyclic diamines increased, more polymers were converted to amic acid salts, decreasing the molecular weight of the copolymers. Therefore, the introduction ratio was limited to a maximum of 20% relative to 6FDA. Furthermore, BAC and BBH were introduced as crosslinkers to prevent the degradation of the mechanical properties of 6T as the content of BAC and BBH increased. Simultaneously, the alicyclic structures in BAC and BBH can suppress the interaction of π electrons, minimizing their impact on the refractive index. The mechanical properties and optical properties of the alicyclic polyimides prepared via copolymerization or crosslinking were studied systemically to demonstrate their potential applications for cover windows in high-curvature displays.

## 2. Materials and Methods

### 2.1. Materials

The chemicals for preparing polyimides, including 6-FDA, TFMB, BBH, and BAC, were purchased from TCI (Tokyo Chemistry Industry, Tokyo, Japan) and used as received without purification. The reaction solvent N-methyl-2-pyrrolidone (NMP, 99.5%) was purchased from Sigma Aldrich (Burlington, MA, USA) and used after removing moisture with a molecular sieve. Acetic anhydride (99.0%) and triethylamine (99.5%) for chemical imidization were purchased from Samchun Chemicals (Seoul, Republic of Korea) and used as received without further purification. Ethanol (EtOH, 99.5%) for re-precipitation was purchased from Samchun Chemicals (Seoul, Republic of Korea) and used as received. The catalysts used in the crosslinking system, including 1-ethyl-3-(3-dimethylaminopropyl)carbodiimide (EDC) and 4-dimethylaminopyridine (DMAP), were purchased from Sigma Aldrich (Burlington, MA, USA) and used as received.

### 2.2. Preparation of Polyimides

#### 2.2.1. Synthesis of 6T-BAC and 6T-BBH Copolymer Series

To prepare polyimides with high flexibility and improved optical properties, 6FDA was used as the anhydride, and TFMB, BBH, and BAC were used as diamines. The composition ratios and polymerization sequence are shown in Table 1 and Figure 1b, respectively. Aliphatic diamines (BAC and BBH) are more basic than aromatic diamines due to the higher electron density of the nitrogen atom. This induces an acid–base reaction with carboxylic acid in the poly(amic acid)(PAA) to form a poly(amic acid) salt, creating strong ionic bonds, which inhibit the condensation reaction and reduce the molecular weight of the copolymer. Since molecular weight plays an important role in the mechanical properties of polyimides, the copolymer was prepared by varying the BAC and BBH contents (5 mol%, 10 mol%, and 20 mol%) relative to 6FDA, with a maximum of 20%. In a glove box under nitrogen atmosphere, BAC (1.576 mmol) and TFMB (14.18 mmol) were dissolved in anhydrous NMP for 30 min in a 250 mL round-bottom flask, followed by adding anhydride 6FDA (15.76 mmol) and stirring at 25 °C for 24 h to obtain a viscous 15 wt% 6T-BAC_PAA solution. Chemical imidization was then performed by adding acetic anhydride (33.72 mmol) and triethylamine (22.54 mmol) to 6T-BAC_PAA, and the mixture was stirred for 12 h to obtain a polyimide varnish. For purification, the varnish was re-precipitated in EtOH to remove unreacted acetic anhydride and triethylamine and then vacuum-dried at 80 °C for 6 h to remove residual EtOH, yielding a white polyimide powder. The 6T-BBH_PAA series was prepared using the same method as described above.

#### 2.2.2. Synthesis of 6T-CL-BAC and 6T-CL-BBH Crosslinked Polymer Series

6T-CL-BAC and 6T-CL-BBH were prepared using 6T as the basic backbone structure and BAC and BBH as crosslinkers. The composition ratios and polymerization sequence are shown in Table 1 and Figure 1a. In a glove box under a nitrogen atmosphere, TFMB (15.76 mmol) and anhydrous NMP were added to a 250 mL round-bottom flask and allowed to dissolve for 30 min. Then, 6FDA (15.76 mmol) was added, and the mixture was stirred at 25 °C for 24 h to obtain 6T_PAA. At this point, an additional 0.03 eq (0.4728 mmol) of TFMB was added, and the reaction proceeded for another 6 h to obtain end-capped 6T_PAA. This procedure was carried out to prevent chain extension caused by the reaction of the poly(amic acid) end-COOH upon the addition of BAC and BBH for the crosslinking reaction.

Chemical imidization was then performed by adding acetic anhydride (33.72 mmol) and triethylamine (22.54 mmol), and the mixture was stirred for 12 h to obtain a polyimide varnish. The varnish was re-precipitated in EtOH to remove unreacted acetic anhydride and triethylamine and vacuum-dried at 80 °C for 6 h to remove residual EtOH. After the above purification, white polyimide powder was obtained, which was then dissolved in DMAc. Then, EDC and DMAP were used as catalysts, and BAC was added as a crosslinking agent. This solution was reacted at room temperature for 40 min to prepare 6T-CL-BAC. 6T-CL-BBH was prepared using BBH as a crosslinker, following the same method described above.

#### 2.2.3. Preparation of Polyimide Films

To fabricate the polyimide films, the prepared polyimide powder was dissolved in DMAc to form a polyimide varnish with a solid content of 15 wt%. The varnish was coated onto a glass substrate and soft-baked in a nitrogen atmosphere oven at 80 °C for 15 min and 150 °C for 30 min, with a ramp rate of 3 °C/min. Then, the 6T-BAC, 6T-BBH, 6T-CL-BAC, and 6T-CL-BBH films were obtained by hard baking at 180 °C for 30 min and 250 °C for 30 min. The prepared polyimide film was placed in deionized water for peeling from the glass plate and dried in a vacuum oven for 12 h to remove moisture from the film, finally resulting in a transparent polyimide film.

### 2.3. Measurements and Characterizations

The structure of the polyimides was analyzed by Fourier transform infrared spectroscopy (FT-IR, PerkinElmer, Waltham, MA, USA) in the range of 4000–650 cm^−1^. The molecular orbital energy of polyimide was calculated using the B3LYP method in the Gaussian 16W program. This allowed us to calculate the HOMO, LUMO, and energy gap. The number average molecular weight (M_n_) and weight average molecular weight (M_w_) was determined by gel permeation chromatography (GPC, Futecs Co. Ltd., Seoul, Republic of Korea) analysis, which was conducted at a concentration of 0.5 wt.% at 40 °C using Tetrahydrofuran (THF) (HPLC grade, Chemicals Duksan Corp., Ansan, Republic of Korea) as the eluent. The viscosity of the polyimides was measured using a small sample adapter (SSA18/13R) of the BROOK FIELD DVE model (BROOK FIELD CO., Ltd., Toronto, ON, Canada), with a viscosity range of 3 to 10,000 cP (MPa∙s) and a measuring volume of 6.67 mL. X-ray diffraction (XRD) was carried out to determine intermolecular distances via Rigaku’s Smart Lab, and the measurement conditions were Cu Kα radiation at 45 kV/200 mA. The thermal resistance of the polyimides was measured by thermogravimetric analysis (TGA-50, Shimadzu, Japan) using a platinum pan under a nitrogen atmosphere; this was carried out at a heating rate of 5 °C/min and within a temperature range from 25 °C to 350 °C. To determine the glass transition temperature (T_g_) of polyimides, differential scanning calorimetry (DSC) measurements (DSC; Exstar 7020, SEIKO, Tokyo, Japan) were performed in the temperature range from 25 °C to 400 °C at a heating rate of 10 °C/min. To verify the mechanical properties, a universal testing machine (UTM, Model 3344, Instron Engineering Corp., Canton, MA, USA) was used at a tensile speed of 20 mm/min. Optical properties were measured in the visible region (400 to 800 nm) using a refractive index/birefringence analyzer (Prism coupler, 2010/M, Metricon, Pennington, NJ, USA). In addition, light transmittance was measured by a UV-Vis spectrophotometer (UV-Vis, Cary 3500, Agilent Technologies, Santa Clara, CA, USA) with a measurement range from 400 to 800 nm. Yellowness was measured with a chromaticity/luminance measurement device (MCPD-3000, Otsuka Electronics Co., Osaka, Japan).

## 3. Results and Discussion

### 3.1. Characterization of Polyimides

#### 3.1.1. GPC and Viscosity of Copolymer Series and CL-Polymer Series

The molecular weights of the 6T, 6T-BBH, and 6T-BBH series were determined by GPC, and the molecular weights and viscosities of the polyimide samples are shown in Table 2. The M_w_ of 6T is 93,000 g/mol, which is the highest compared to other polyimides. Aliphatic diamines have a higher electron density of N_2_ compared to aromatic diamines, which makes them strongly basic. This causes an acid–base reaction with the carboxylic acid of the poly(amic acid) to produce poly(amic acid) salts. Therefore, increasing the ratio of BAC and BBH resulted in a slightly lower molecular weight compared to 6T. The CL series (6T-CL-BAC and 6T-CL-BBH) underwent a reprecipitation process for GPC measurements, but due to the formation of a crosslinked network, the samples could not be redissolved in THF—the solvent used for GPC—making it possible to determine their molecular weights.

The average viscosity of the 6T-BAC and 6T-BBH series at 25 °C was 2500 cp, lower than that of 6T (4300 cp). Upon the introduction of BAC and BBH, the proportion of aromatic structures decreases, leading to a decrease in π–π stacking and weaker intermolecular interactions. On the other hand, the viscosity of the CL series is higher (4830 cp), possibly because the crosslinked structures reduced the intermolecular distance and enhanced the intermolecular attraction.

#### 3.1.2. FT-IR Characterization

Figure 1a represents the polyimide films, and Figure 1b,c show the FT-IR spectra of nine polyimides. All samples showed absorption peaks corresponding to the asymmetric stretching and symmetric stretching of the imide ring at 1784 cm^−1^ and 1726 cm^−1^, respectively, as well as the stretching vibration absorption peaks of C-N near 1370 cm^−1^ and 720 cm^−1^ [34], confirming the successful polymerization of the polyimide. For the 6T-BAC and 6T-BBH series, an additional aliphatic C-H stretching absorption peak appeared at 2900 cm^−1^, indicating that the BAC- and BBH-containing monocyclic and bicyclic structures were successfully copolymerized.

Similarly, for 6T-CL-BAC and 6T-CL-BBH, an aliphatic C-H stretching peak at 2900 cm^−1^ was observed, suggesting the presence of BAC and BBH. Furthermore, the C=O stretching absorption peak of the amide (1672 cm^−1^) and the N-H bending absorption peak of the amide (1535 cm^−1^) confirmed the successful formation of the crosslinked structure [35,36].

#### 3.1.3. Density Functional Theory

The HOMO and LUMO energy levels of the copolymer series and crosslinked (CL) polymer series were analyzed by Gaussian-based DFT calculations, and the electron density distributions are shown in Figure 2.

The CTC structure between polyimide molecular chains is formed by electron transfer between diamine, an electron donor, and dianhydride, an electron acceptor. In general, the narrower the HOMO-LUMO energy gap, the easier the electron transfer, which promotes the formation of CTCs; this tends to decrease the light transmittance and increase the refractive index [37]. The 6T-BAC and 6T-BBH series exhibited an increase in HOMO energy due to the introduction of cyclic and heterocyclic structures, but the overall energy gap was characterized by a wider gap compared to 6T. This is interpreted to mean that the π-electron cloud is less extended by the introduced structures than in 6T, and the electron density is concentrated in a narrower region, resulting in a relatively higher HOMO energy. These calculation results also tend to agree with other experimental results in the text.

On the other hand, the electron density of the 6T-CL-BAC and 6T-CL-BBH series with the introduced crosslinking system was analyzed (see Figure 2), and it was found that the HOMO and LUMO electrons were scarcely distributed in the crosslinked region. This suggests that the introduction of alicyclic crosslinking systems has a very low impact on the refractive index and optical transparency of polyimide films. The introduction of alicyclic crosslinking systems is expected to be a way to effectively increase mechanical properties with a minimal loss of optical properties in the molecular design of optical materials.

In addition, the 6T-CL-BAC and 6T-CL-BBH series showed a slight decrease in the HOMO-LUMO energy gap due to the formation of new covalent bonds. However, these results are based on an idealized model assuming a crosslinking ratio of 100%, and it is believed that in actual fabricated films, the crosslinking ratio is about 2%, which has a very limited effect on the overall electronic structure.

The Gaussian calculation results correspond to the HOMO and LUMO energy levels of individual molecules and thus have limitations in directly explaining intermolecular charge transfer complex (CTC) formation. Nevertheless, the calculated energy levels exhibit trends that align well with the optical properties discussed in the text, such as the refractive index, UV-Vis spectra, and yellow index, as well as the changes in intermolecular distances observed by XRD. These results suggest that, while not conclusive, the calculations can serve as a useful indirect guideline.

#### 3.1.4. X-Ray Diffraction Data Analysis

XRD measurements were conducted to analyze the intermolecular distances of polyimides incorporating either copolymer or crosslinking systems, as shown in Figure 3. All polyimide samples exhibited amorphous structures.

In the case of the 6T-BAC and 6T-BBH copolymer series, the main diffraction peak positions shifted to lower 2θ values compared to the conventional 6T, and the corresponding d-spacing values showed a tendency to increase. This trend is attributed to the incorporation of alicyclic structures (BAC and BBH) into the polymer backbone, which increases chain flexibility and disrupts π–π stacking, thereby expanding the intermolecular distances. Notably, BAC, with its simple alicyclic structure, imparts greater flexibility than the bridged bicyclic BBH, resulting in the 6T-BAC series exhibiting larger d-spacing values than the 6T-BBH series. These findings suggest that the suppression of π–π interactions in the copolymer series contributes to improved optical transparency, while enhanced backbone flexibility facilitates the fabrication of more flexible polyimide films.

In addition, the 6T-CL-BAC and 6T-CL-BBH crosslinked series exhibited weak additional diffraction peaks near 8°, indicating the formation of crosslinked structures [38]. At the same time, diffraction peaks corresponding to the original 6T-based structure were still observed, suggesting that the overall molecular arrangement of the polyimide was preserved. This implies that the introduction of alicyclic structures into the crosslinked system does not compromise optical properties, as the original molecular ordering is maintained, while the formation of crosslinked networks contributes to improved mechanical properties.

### 3.2. Thermal Properties Analysis

The heat resistance results of the fabricated polyimide films are shown in Figure 4. All polyimide films exhibited high heat resistance, with only 5% weight loss at temperatures above 450 °C. Aliphatic structures (BAC and BBH) have lower thermal resistance compared to aromatic structures. Therefore, compared to 6T, the thermal decomposition temperature tended to decrease gradually as the aliphatic ratio increased in the 6T-BAC and 6T-BBH series. [39,40] In particular, the thermal decomposition temperature of the 6T-BAC series was approximately 474 °C, which was lower than that of the 6T-BBH series (T_d5_: approximately 503 °C) due to the high rigidity of the bridged heterocyclic structure in BBH.

The thermal decomposition temperatures of the 6T-CL-BAC and 6T-CL-BBH series were approximately 490 °C and 512 °C, respectively, which are lower than that of 6T. This can be explained by the following three structural and chemical factors. First, structural changes occur due to the formation of amide bonds, which partially replace the original imide ring structures. While imide rings exhibit excellent thermal stability at high temperatures, amide bonds are relatively less stable, and thus, this structural conversion may contribute to a reduction in the overall thermal decomposition temperature. Second, the introduction of thermally less stable crosslinkers occurs. BAC and BBH are both saturated alicyclic compounds, and their structures are based on single C-C bonds. Such structures are thermally less stable than aromatic systems. As a result, polymer systems incorporating these crosslinkers may exhibit lower thermal decomposition temperatures than fully aromatic systems like 6T. Third, a decrease in intermolecular interactions occurs due to the introduction of the crosslinking system. While 6T possesses strong chain–chain interaction via π–π stacking between aromatic moieties, the CL series contains alicyclic diamines that reduce such interactions. This leads to an increase in intermolecular distance, an expansion of free volume, and enhanced chain mobility, which may lower the onset temperature of decomposition. This interpretation is also consistent with the XRD results showing increased interchain spacing in the CL series.

However, the influence of reduced intermolecular interactions on thermal decomposition temperature should be considered an indirect factor. The primary decomposition temperature should be considered a direct factor. The primary causes are identified as the formation of amide structures and the incorporation of thermally less stable crosslinkers.

DSC measurements were performed to analyze the thermal properties of the as-prepared polyimide. The corresponding graphs are shown in Figure 5, and T_g_ and T_d_ are shown in Table 3. The T_g_ of 6T is approximately 328 °C, which is the highest among all samples. The π–π stacking of the aromatic structure in 6T causes narrow chain spacing and strong intermolecular interactions, increasing T_g_ and limiting chain mobility. In contrast, the 6T-BAC and 6T-BBH series, which contain alicyclic ring structures, exhibited lower T_g_ values of approximately 292 °C and 308 °C, respectively. The low T_g_ is related to the increased chain spacing due to the bent structure of the molecules and the weakened intermolecular interactions due to the alicyclic structure. In addition, as the ratio of BAC and BBH increased, the irregularity of the chain increased, resulting in a gradual decrease in T_g_.

The T_g_ values of 6T-CL-BAC and 6T-CL-BBH were approximately 297 °C and 314 °C, respectively, which are lower than that of 6T but higher than those of 6T-BAC and 6T-BBH. According to previous studies, when aromatic diamines are used as crosslinking agents, strong π–π interactions and enhanced chain rigidity are typically induced, resulting in an increased T_g_ [41,42]. However, in this study, despite the introduction of a crosslinking system, a decrease in T_g_ was observed. Based on the DFT results, it was theoretically confirmed that there was almost no electron density distributed at the crosslinked sites, indicating that π–π interactions were not formed. Furthermore, the XRD analysis showed an increase in intermolecular spacing compared to the original 6T upon crosslinking. This can be attributed to the fact that the crosslinking agents used in this study were structurally flexible alicyclic diamines, which are incapable of inducing π–π interactions. As a result, even with the introduction of a crosslinking system, a decrease in T_g_ was observed.

### 3.3. Optical Properties of Polyimide Films

#### 3.3.1. UV-Vis

The light transmittance of all polyimide films was evaluated by UV-Vis spectral analysis in the visible light region (400 nm to 800 nm), which is shown in Figure 6 and Table 4. For the 6T-BAC and 6T-BBH series, the maximum transmittance was 92%, and the transmittance at 400 nm was approximately 82%, which was not a significant change compared to 6T. The light transmittance of polyimides is primarily determined by the conjugated structure of the molecules and the intermolecular π–π stacking. An expanded conjugated π electron system facilitates electronic transitions and increases the molar extinction coefficient. Likewise, stronger π–π interactions increase intermolecular electron transfer, resulting in a higher molar extinction coefficient. In general, polyimides form CTCs with stronger π–π electron interactions, which increases the molar extinction coefficient. Thus, according to the Lambert–Beer law, the absorbance of polyimides increases, resulting in lower transmittance. In addition, the stronger the π–π electron interactions between the aromatic structures of the polyimide, the denser the intermolecular packing, which induces internal scattering and further reduces transmittance [43,44].

In this study, 6T-BAC and 6T-BBH reduce the planar structure of the polymer and weaken charge transfers due to their alicyclic and bent structures, leading to a reduction in CTC structure and higher permeability. As a bridged heterocyclic structure, BBH exhibits a more symmetrical chain structure compared to BAC, leading to relatively more regular intermolecular packing, which may explain the lower transmittance of 6T-BBH.

The maximum transmittance of 6T-CL-BAC and 6T-CL-BBH was 92%, and the transmittance at 400 nm was approximately 80%, showing no significant difference compared to 6T. As previously described, transmittance is primarily influenced by intermolecular π–π interactions. Since the BAC and BBH used as crosslinkers have alicyclic structures that make π–π interactions difficult to induce and increase the intermolecular distance, they are unable to form densely packed arrangements, which likely explains the minimal difference in transmittance compared to 6T.

Also, in the case of the copolymer series, BAC and BBH were introduced as part of the backbone, providing greater flexibility in molecular packing compared to the crosslinked series. As a result, π–π interactions were more effectively suppressed. This structural difference aligns well with the increase in intermolecular distance observed in the XRD analysis, and it can serve as an interpretation for the higher maximum transmittance and transmittance at 400 nm exhibited by the copolymer series compared to the crosslinked series.

#### 3.3.2. Refractive Index and Yellow Index (YI)

The refractive index of the polyimide films measured with a refractive index/birefringence analyzer is shown in Figure 7 and Table 4. The refractive index, which is the ratio of the speed of light in a vacuum to the speed of light in a material, is a measure of how much light slows down as it passes through a material. This difference in speed is caused by light interacting with electrons in a material. 6T has a refractive index of about 1.57, lower than the typical refractive index of polyimide (1.7), which can be related to the low polarization of the 6T molecules. The CF_3_ substituents in 6T have low polarizability, which weakens the interactions between light and the electrons in the material, resulting in a reduced refractive index. The refractive indices of the 6T-BAC and 6T-BBH series were lower than that of 6T (1.531 and 1.546, respectively) [45]. The refractive index tended to decrease as the alicyclic ratio increased. BAC and BBH with alicyclic structures can form a bend in the molecule, which increases the intermolecular distance. This results in fewer interactions between light and molecules within the same volume, which reduces the refractive index. In addition, 6T-BBH has a higher refractive index than 6T-BAC because of its bridged heterocyclic structure, which reduces intermolecular distances and enhances the interaction between light and molecules within the same volume.

6T-CL-BAC and 6T-CL-BBH have refractive indices of 1.545 and 1.551, respectively, confirming a decrease in refractive index compared to the original 6T. Based on the previous XRD results, new peaks appeared when the crosslinked structure was introduced, indicating a slight increase in intermolecular distance. This can be explained by the fact that as the intermolecular distance increases, the electron density in the same volume decreases, resulting in a weaker interaction with light and hence a lower refractive index. Furthermore, according to the Gaussian-based electron density analysis, HOMO/LUMO electrons are rarely distributed in the crosslinked sites, suggesting that the crosslinking structure has a limited effect on the optical properties.

In addition, all polyimide films exhibited high transparency with a YI of about 1.2, the figures of which are shown in Figure 7 and Table 4. In particular, the transparency of 6T-CL-BAC and 6T-CL-BBH was high, despite the introduction of a crosslinking system, due to the use of an alicyclic structure as a crosslinking agent. As a result, all of the fabricated polyimide films achieved a refractive index close to 1.5, which is lower than that of the conventional 6T.

**Table 4 polymers-17-02081-t004:** Optical properties of 6T-BAC and 6T-BBH series and 6T-CL-BAC and 6T-CL-BBH.

Polyimide	Thickness (μm)	Refractive Index (632.8 nm)	T% (400 nm)	T% Max	YI
6T	25	1.568	87.79	94.02	1.30
6T-BAC-1	26	1.550	88.28	95.70	1.18
6T-BAC-2	24	1.542	89.87	97.00	1.13
6T-BAC-3	25	1.531	90.19	97.25	1.08
6T-BBH-1	23	1.558	87.24	95.52	1.20
6T-BBH-2	25	1.549	89.33	95.79	1.16
6T-BBH-3	26	1.546	88.84	96.08	1.13
6T-CL-BAC	27	1.545	86.73	94.33	1.25
6T-CL-BBH	26	1.546	86.01	94.66	1.28

### 3.4. Mechanical Properties of Polyimide Films

The mechanical properties of the polyimide films are shown in Figure 8 and Figure 9. The tensile strengths of 6T-BAC and 6T-BBH series were measured to be 63~83 MPa and 72~84 MPa, respectively, lower than that of 6T (approximately 87 MPa). The tensile strength decreased as the content of BAC and BBH increased [45]. The introduction of alicyclic structures generally tends to increase the chain mobility and free volume of the polymer, which in turn leads to a decrease in tensile strength. However, BBH exhibited higher tensile strength because of its bridged heterocyclic structure, which showed greater stiffness than BAC. Meanwhile, the fracture strain of 6T was approximately 4.4%, while the 6T-BAC and 6T-BBH series had relatively high flexibility, ranging from 6.41 to 18.67% and 4.87 to 14.38%, respectively. This is likely because the introduction of alicyclic structures (BAC and BBH) relaxes intermolecular interactions and increases the free volume.

The 6T-CL-BAC and 6T-CL-BBH series with crosslinked structures showed increased mechanical strength and flexibility compared to 6T, with tensile strengths of 91.55 MPa and 93.65 MPa, respectively, and fracture strain values of 11.06% and 9.77%, respectively. Crosslinking systems are generally known to improve mechanical strength by increasing interchain density while reducing flexibility. However, in the case of 6T-CL-BAC and 6T-CL-BBH, not only do they possess flexible alicyclic-based structures but they also have a low degree of crosslinking, which helps maintain chain mobility. In addition, their intermolecular distance is increased compared to 6T, which reflects an increase in strain. Nevertheless, both the copolymer series (6T-BAC and 6T-BBH) and the crosslinked series (6T-CL-BAC and 6T-CL-BBH) demonstrated improved flexibility compared to the original 6T, indicating their potential suitability for flexible cover windows requiring a small bending radius.

## 4. Conclusions

In this study, flexible polyimide films suitable for display cover window applications were developed by introducing alicyclic diamines—1,3-bis(aminomethyl)cyclohexane (BAC) with a monocyclic structure and bis(aminomethyl)bicyclo[2,2,1]heptane (BBH) with a bicyclic structure—into a 6FDA-TFMB-based polyimide system via either copolymerization or crosslinking.

The resulting copolymers (6T-BAC and 6T-BBH) exhibited excellent optical properties, with refractive indices of approximately 1.53–1.54, maximum transmittance exceeding 95%, and yellow indices around 1.1—superior to those of 6T. Their glass transition temperatures ranged from 280 to 320 °C, slightly lower than that of 6T (328.5 °C), and their tensile strengths (63–80 MPa) were also reduced. However, all copolymer films demonstrated strain-at-break values above 4.8%, indicating a favorable balance between optical clarity and mechanical flexibility. Notably, the 6T-BAC series exhibited the most desirable combination of optical and flexibility.

On the other hand, the crosslinked 6T-CL series maintained high optical transparency (RI ≈ 1.54; transmittance ≈ 94%; yellow index of 1.25–1.28) while achieving significantly improved mechanical properties, with tensile strength values of 91–93 MPa and strain-at-break values ranging from 9% to 11%. While the incorporation of alicyclic structures through copolymerization enhanced optical clarity and flexibility, it also led to diminished mechanical strength. In contrast, the introduction of a crosslinking system allowed these films to retain their optical and flexible properties while simultaneously reinforcing their mechanical strength.

Taken together, these findings demonstrate that the alicyclic polyimide structures designed in this study satisfy the combined requirements for optical clarity, flexibility, and mechanical durability, making them promising candidates for display cover windows with small curvature radii. Furthermore, by systematically examining the relationship between structural design and properties based on the method of introducing alicyclic diamines, this work offers practical guidance for the development of polymer architectures tailored to advanced display technologies.

## Data Availability

The data presented in this study are available upon request from the corresponding author.

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
