# Peer review of "Transparent Alicyclic Polyimides Prepared via Copolymerization or Crosslinking: Enhanced Flexibility and Optical Properties for Flexible Display Cover Windows"

_polymers, 2025, doi:10.3390/polym17152081_

Round 1
Reviewer 1 Report
Comments and Suggestions for Authors
Comments for Manuscript ID: polymers-3779712
The manuscript describes the synthesis and characterization of a series of alicyclic polyimides by using 4,4'-(hexafluoroisopropylidene)diphthalic anhydride and 2,2'- bis(trifluoromethyl)benzidine as the base monomers of polyimides. Additionally, 1,3-bis(aminomethyl)cyclohexane and bis(aminomethyl)bicyclo[2,2,1]heptane were introduced as co-monomers or crosslinking agents to the initial polyimide. The optical, mechanical, and thermal properties of the obtained polyimide, copolymers and crosslinked polymers were compared. The data suggest their potential use as materials for flexible display cover windows, as indirect input from the measurements.
In the current form, the manuscript cannot be published in Polymers and a series of modifications must be performed by the authors, as follows:
- There are mistakes (e.g. peaks in FTIR instead of absorption band, Mw or Mn instead of Mn and Mw, etc.), grammatical errors and typos along the manuscript, which should be revised and corrected.
- The authors don’t offer enough details regarding the synthesis of the co(polymers) or the film formation (solvent, solution concentration) in the experimental part. Thus, the results can’t be reproducible based on the details given in the methods section. Also, I suggest to add in Scheme 1 the codes of the corresponding (co)polymer(s) and the condition of reactions on the arrows for a better understanding.
- There is a misunderstanding in the text regarding the addition of 30 and 50 wt% of BAC and BBH in polyimide. The authors claim that: “Since molecular weight plays an important role in the mechanical properties of polyimides, the co-polymer was prepared by varying the BAC and BBH contents (5%, 10%, and 20%) relative to 6FDA, with a maximum of 20%”. Since the only investigated properties of 6T-BAC (30% and 50%) and 6T-BBH (30% and 50%) are the molecular weight characteristics and viscosities and no other comments were made on them along the manuscript, I don’t see the reason for their incorporation.
- In the thermal properties section, the last paragraph:” However, the influence of reduced intermolecular interactions on thermal decomposition temperature should be considered an indirect factor. The primary decomposition temperature should be considered an indirect factor. The primary causes are identified as the formation of amide structures and the incorporation of thermally less stable crosslinkers.” should be reformulated.
- There is a mistake in the numbering of the figures in section 3.2 (line 252) which leads to an inconsistence in the further numbering of the figures in the manuscript.
Comments on the Quality of English Language
In general, the quality of English language is suitable for the journal.
Reviewer 2 Report
Comments and Suggestions for Authors
The manuscript presents interesting developments in transparent and flexible polyimide materials; however, several important issues should be addressed to improve the scientific clarity and consistency of the work.
The claim that the materials achieved "a refractive index of 1.5" and are suitable for next-generation flexible display applications requires stronger support. Additionally, the phrase “great potential” is vague.
There is considerable inconsistency and disorder in the figure numbering and their references within the text. Figures appear out of order, and their in-text mentions are sometimes missing or mismatched. This needs to be thoroughly revised for readability and accuracy.
The presentation of thermal stability data would be significantly improved by including TGA results in differential form (DTG curves). This would allow for more precise identification of decomposition steps and better comparison between materials, particularly for evaluating the effects of crosslinking or different diamine structures.
If the synthesized polyimide materials are fully amorphous, as the XRD results and structural features suggest, this should be explicitly stated in the manuscript. Clear identification of the polymer morphology is essential, especially in the context of optical clarity and mechanical flexibility, both of which are often linked to amorphous character.
Accurate analysis of optical properties, particularly refractive index, can be challenging, as measurement errors are common, especially in highly transparent systems. While the authors employed a refractive index/birefringence analyzer suitable for such materials, the use of ellipsometry, as described in the referenced work https://doi.org/10.1364/AO.54.006208, could offer deeper insights into optical dispersion and film structure, particularly under low optical contrast conditions. Naturally, this would require the preparation of thinner films. Given the subtle structural variations introduced through alicyclic substitution and crosslinking, ellipsometry would serve as a valuable complementary technique to the current optical characterization. At minimum, a discussion of this possibility, supported by the cited reference, should be included in the manuscript.
Reviewer 3 Report
Comments and Suggestions for Authors
In this manuscript, the author improveed the flexibility of aromatic polyimides films via co-polymerization or crosslinking with alicyclic polyimides, which is meaningful to flexible display. However, the whole manuscript was in a bad organizing and writing, the following issues should be addressed before acceptance:
- The format should revised carefrully corresponding to the template.
- In the Abstract, the author stated “Furthermore, the refractive index of most transparent polyimides is approximately 1.57, which differs from that of the optically clear adhesive (OCA) and window materials, typically around 1.5, resulting in visual distortion. ”, however, this issue was not addressed via co-polymerization or crosslinking with alicyclic polyimides, as RI of the obtained transparent films are around 1.55. Thus, this sentence should deleted.
- Table 1 should located within section 2.2.1, and scheme 1 should mentioned within the text.
- Scheme 1 should revised, the abbreviations of raw materials should added, and the raw material of alicyclic diamine were missed within (a).
- Section 2.2.3, how can you applied the prepared polyimide onto a glass plate? as the obtained 6T-BAC and 6T-BBH co-polymers are white powder (Section 2.2.1).
- The pictures of all prepared transparent polyimide films should provided (9 samples).
- The caption of all Figures should located after the Figures.
- The resolution of the FT-IR spectra should improved.
- There was almost no references in the section of Results and Discussion, how can you support your analysis and judge your results?
- In the Conclusions, the author stated “In particular, 6T-CL-BAC exhibited excellent overall performance, showing a refractive index close to 1.5 and a high light transmittance of over 90%, thereby achieving both desirable physicaland optical properties”, however, the RI of 6T-CL-BAC is 1.545, only 0.023 lower than 6T (1.568) and much higher than 1.5. Thus, this sentence should deleted.
- Other important value data should summarized within the Conclusions.

Reviewer 4 Report
Comments and Suggestions for Authors
In this work, transparent alicyclic polyimide materials were prepared through copolymerization or crosslinking, and their properties were characterized using various methods. Based on the current manuscript, there are some issues that need to be addressed.
- The author prepared (6T-BAC and 6T-BBH series) and (6T-CL-BACand 6T-CL-BBH series). In the abstract, it was not highlighted which material was better?
- Has there been any similar report on the modification method or monomer selected by the author to modify polyimide to increase its flexibility and reduce its refractive index? In the introduction, the author needs to provide examples and comparative explanations of the relevant studies of predecessors.
- The annotation in Figure 1(a) is not clear, and its resolution needs to be improved.
- The author stated that the C=O stretching peak and N-H bending peak of the amide confirmed the successful formation of the cross-linked structure. How did the author determine its cross-linked structure based on the infrared peak?
- These calculation results also tend to agree with other experimental results in the text. The "other results" referred to here need to be clarified.
- The author conducted theoretical calculations on 6T-copolymer series & 6T-CL series based on density functional theory. Should the author first carry out theoretical calculations and then determine the materials with better effects for experiments? This way, it avoids preparing so many materials for research. What is the basis for the author's design in this way?
- The characterization and analysis of polyimide in the manuscript should appropriately increase corresponding citations, which will make the result analysis more convincing.
- The data results presented in sections 3.3.1 and 3.3.2 should be compared with those from similar literature to highlight the advantages of this work.
- The comparison of the prepared 6T-BAC and 6T-BBH series in this manuscript should not be limited to their performance compared to 6T. The differences between the 6T-BAC and 6T-BBH series materials also need to be compared and explained.
Round 2
Reviewer 2 Report
Comments and Suggestions for Authors
The authors have answered to all my issues and the paper can be accepted in its present form.
Reviewer 3 Report
Comments and Suggestions for Authors
Corrections were performed, the manuscript has been improved and now it is suitable for publishing
Reviewer 4 Report
Comments and Suggestions for Authors
The author has resolved the issue I raised, and the current version is acceptable.